# Pretraining Deformable Image Registration Networks with Random Images

**Junyu Chen**[1]  🆔                                                    JCHEN245@JHMI.EDU
[1] *Department of Radiology and Radiological Science, Johns Hopkins University*
**Shuwen Wei**[2]  🆔                                                    SWEI14@JHU.EDU
[2] *Department of Electrical and Computer Engineering, Johns Hopkins University*
**Yihao Liu**[3]  🆔                                              YIHAO.LIU@VANDERBILT.EDU
[3] *Department of Electrical and Computer Engineering, Vanderbilt University*
**Aaron Carass**[2]  🆔                                           AARON_CARASS@JHU.EDU
**Yong Du**[1]  🆔                                                      DUYONG@JHMI.EDU

**Editors:** Accepted for publication at MIDL 2025

## Abstract

Recent advances in deep learning-based medical image registration have shown that training deep neural networks (DNNs) does not necessarily require medical images. Previous work showed that DNNs trained on randomly generated images with carefully designed noise and contrast properties can still generalize well to unseen medical data. Building on this insight, we propose using registration between random images as a proxy task for pretraining a foundation model for image registration. Empirical results show that our pretraining strategy improves registration accuracy, reduces the amount of domain-specific data needed to achieve competitive performance, and accelerates convergence during downstream training, thereby enhancing computational efficiency. Our implementation is available at https://bit.ly/3GioAK8.

**Keywords:** Image Registration, Self-supervised Learning, Pretraining

## 1. Introduction

Medical image registration has advanced significantly with the rise of deep learning, particularly through recent unsupervised methods (Chen et al., 2024). Unlike traditional optimization-based approaches that perform pairwise optimization, unsupervised deep learning methods optimize a global objective across a dataset, enabling the trained network to generalize to unseen images at inference. However, this generalization is inherently constrained by the size of the dataset and the duration of training.

A distinctive property of deep learning-based registration is that training images do not need to be drawn from the same distribution as test images. Hoffmann et al. (2021) demonstrated that even randomly generated images can be used to train a registration network that performs effectively on brain magnetic resonance imaging (MRI), although it is never exposed to such images during training. They also showed that synthetic images resembling brain MRI can further improve performance. However, generating such synthetic images is a meticulous process that requires careful tuning of noise, resolution, and contrast to match the characteristics of downstream medical images.

In this work, we propose a proxy pretraining task based on random image registration to train a foundation registration model that can be fine-tuned for downstream tasks. We show empirically that this pretraining strategy improves training efficiency, accelerates convergence, and enhances performance when domain-specific data is limited.

## 2. Methods

Let $f, m : \Omega \to \mathbb{R}$ be the fixed and moving images, respectively, defined in a 3D domain $\Omega \subseteq \mathbb{R}^3$. A DNN $g$ predicts a deformation field $\phi = g(f, m)$ that warps $m$ toward $f$. An overview of the proposed pretraining strategy is illustrated in Figure 1 in Appx. B.

**Lightweight Decoder.** Inspired by recent pretraining strategies for vision foundation models (He et al., 2022; Kirillov et al., 2023; Oquab et al., 2023), we propose pretraining only the encoder component of a registration DNN. Specifically, we employ an asymmetric architecture by attaching lightweight temporary decoders to the encoder during pretraining. Unlike typical reconstruction-based pretraining, our approach directly optimizes the registration loss between warped moving and fixed images. Each lightweight decoder, consisting of two convolutional layers and a final convolutional "registration head," produces a deformation field. These decoders are attached to each resolution stage of the encoder, allowing effective multi-resolution feature learning. After pretraining, the decoders are discarded, and only the encoder is retained for subsequent registration tasks.

**Self-distillation.** We further introduce an ensemble-based self-distillation strategy (Zhang et al., 2021) to enhance pretraining. Each decoder predicts a Gaussian distribution over the deformation field, parameterized by its mean and variance. An ensemble distribution is also computed for all decoders, and the Kullback–Leibler (KL) divergence between each individual decoder's distribution and the ensemble distribution is minimized to encourage consistency between the encoder stages and improve representation learning.

**Pretraining Proxy Task.** We propose to use the registration of randomly generated image pairs as our pretraining task. Following (Hoffmann et al., 2021), a multi-channel Perlin noise (Perlin, 2002) is generated, and an `argmax` collapses these channels into an image with distinct random shapes, each assigned a constant intensity. Unlike (Hoffmann et al., 2021), we omit additional Gaussian noise and bias fields. Diffeomorphic deformation fields, generated from three-channel Perlin noise, deform these random images to form registration pairs. Details of this process are provided in Appx. C. This random image data is generated efficiently on-the-fly during training, providing virtually infinite image pairs.

The overall loss function for pretraining is defined as follows:

$$\begin{aligned}
\mathcal{L}_{pretrain}(m, f) & = \mathcal{L}_{NCC}(m \circ \mu_{\phi_{ens}}, f) + \lambda \|\nabla u_{ens}\|^2 + \\
& \eta \sum_{k \in K} \frac{1}{K} \mathcal{D}_{KL} \left( \mathcal{N}(\mu_{\phi_{ens}}, \sigma^2_{\phi_{ens}}) \| \mathcal{N}(\mu_{\phi_k}, \sigma^2_{\phi_k}) \right),
\end{aligned} \tag{1}$$

where $\mathcal{L}_{NCC}$ is the normalized cross-correlation (NCC) loss, the ensemble deformation field is represented by the mean $\mu_{\phi_{ens}}$ and variance $\sigma^2_{\phi_{ens}}$, while $\mu_{\phi_k}$ and $\sigma^2_{\phi_k}$ denote the estimates from the decoder at stage $k$. The displacement field $u_{ens}$ is derived from $\mu_{\phi_{ens}}$, and $K$ is the number of encoder stages. The hyperparameters $\lambda = 1$ and $\eta = 1e - 7$ are set empirically. After pretraining, the decoders are discarded and the encoder weights are transferred to the backbone registration network for fine-tuning, with all layers remaining learnable.

## 3. Results and Conclusion

**Experimental Setup.** We evaluated the proposed pretraining strategy on a downstream brain MRI registration task using the IXI dataset. TransMorph (Chen et al., 2022b,a) served

as the backbone, with its encoder extracted for pretraining. Details on the pretraining setup are provided in Appx. A. The baseline methods include: deedsBCV (Heinrich et al., 2013), SynthMorph (Hoffmann et al., 2021), and ConvexAdam (Siebert et al., 2024).

**Ablation Studies.** We conducted ablation studies to evaluate the use of Dice loss versus NCC loss in Equation (1), as originally used in (Hoffmann et al., 2021), and the impact of including or omitting KL loss. Results in Appx. D show that Dice loss is suboptimal compared to NCC loss, although it still improves upon the baseline (no pretraining) in both training loss and validation Dice score—likely due to a mismatch in similarity measures, as NCC loss is also used during fine-tuning for downstream task. Including KL loss reduces training loss (see Figure 5) but slightly lowers validation Dice scores, suggesting that the training objective does not align perfectly with the validation metric. Table 1 further shows that training the full registration model solely on random images without fine-tuning (as in SynthMorph-style training) yields subpar performance, highlighting the necessity of fine-tuning for downstream tasks. Additionally, full-model pretraining is also less efficient, taking 0.022 min/image pair compared to 0.012 min/image pair for encoder-only pretraining.

**Reducing Data Requirements.** We further show that our pretraining strategy reduces the amount of labeled data needed to achieve competitive performance. We trained models using 80%, 60%, 40%, 20%, 10%, and 5% of the IXI dataset (322, 242, 161, 81, 40, and 20 images, respectively). As shown in Figure 6 and Appx. E, pretraining enables the model to achieve similar performance using just 5% of the data as a model trained on 40% without pretraining. Likewise, Table 2 shows that the use of only 10% of the data with pretraining matches the performance of training on 100% without pretraining, thus demonstrating the effectiveness of the method in low-data regimes.

We also compared pretraining with random images to using in-domain brain MRIs from another dataset (Nugent et al., 2022) preprocessed similarly. As shown in Figure 7, random image pretraining yields lower training loss and better validation Dice in early epochs, while brain-image pretraining eventually slightly surpasses it in both validation and test Dice (see Table 3). These results suggest that using registration of random images as a proxy task can improve generalizability without requiring any in-domain data.

**Comparison with Existing Methods.** We compared our approach against widely used baselines. As shown in Table 4, TransMorph pretrained using our method achieves the highest performance among all evaluated methods while maintaining smooth deformations through time-stationary velocity fields. Additional lung registration results in Appx. H further demonstrate the effectiveness of the method in data-limited scenarios.

**Conclusion.** We proposed a pretraining strategy for registration foundation models using random image registration as a proxy task. Experimental results demonstrate improved generalizability and performance. This approach may benefit large-scale training by reducing the number of epochs needed for convergence and is also effective for fine-tuning in scenarios with limited domain-specific training data.

**Acknowledgment.** This work is supported by grants from the National Institutes of Health (NIH), United States, R01CA297470, R01EB031023, U01EB031798, and P01CA272222.

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

## Appendix A. Detailed Experimental Setup

For pretraining, 3,000 random image pairs were generated per epoch, and models were trained for 50 epochs. For the downstream registration task, all models were trained for 250 epochs. The downstream registration task involves atlas-to-subject registration on T1-weighted MRIs using the IXI dataset. The fine-tuning loss combines $\mathcal{L}_{NCC}$ and a diffusion regularizer, both weighted equally (i.e., $\lambda = 1$). The IXI dataset went through preprocessing steps that included skull stripping, affine alignment, bias field correction, and intensity normalization using FreeSurfer (Fischl, 2012). The dataset was divided into training, validation, and testing sets in a 7:1:2 ratio, resulting in 403, 58, and 115 images, respectively. Both pretraining and fine-tuning used the Adam optimizer, with learning rates of $4 \times 10^{-4}$ and $1 \times 10^{-4}$, respectively. All training and experiments were conducted on an NVIDIA H100 GPU.

Registration accuracy was evaluated using Dice to measure the segmentation overlap between the warped moving and fixed images, while the smoothness of the deformation was assessed using the percentage of non-diffeomorphic volume (NDV) (Liu et al., 2024).

## Appendix B. Overview of the Proposed Pretraining Strategy

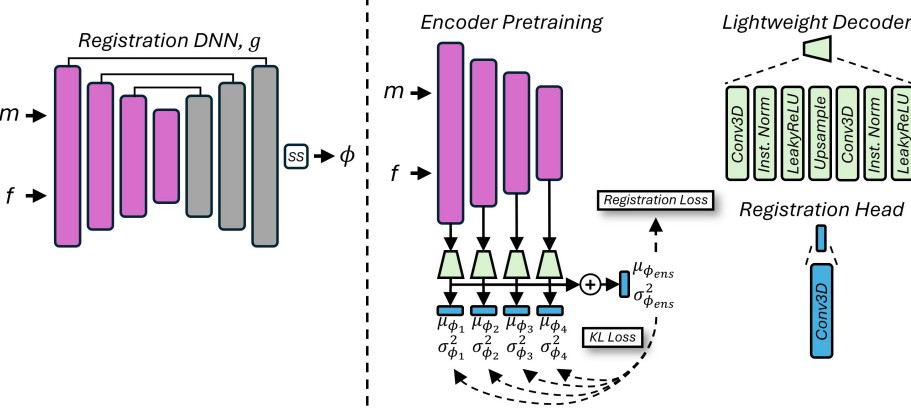

Figure 1: Overview of the proposed pretraining strategy. "SS" denotes scaling-and-squaring, used to generate a deformation field from a time-stationary velocity field (Ashburner, 2007). The encoder of the registration network is extracted and paired with a series of lightweight decoders, each followed by a registration head, attached at different stages of the encoder to estimate deformation fields. An ensemble deformation field is also computed by aggregating feature maps from all decoders. The use of simplistic decoders encourages the encoder to take greater responsibility for learning meaningful representations for image registration.

## Appendix C. The Process of Generating Random Image Pairs

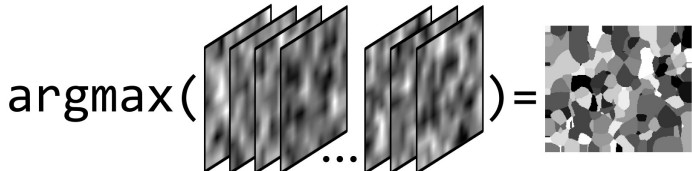

Figure 2: Visualization of the random image generation process with randomly shapes. A multi-channel Perlin noise (Perlin, 2002) is first generated, where each channel is an unique realization using a different random seed. The argmax operation is then applied across channels to collapse them into a single image composed of distinct random shapes.

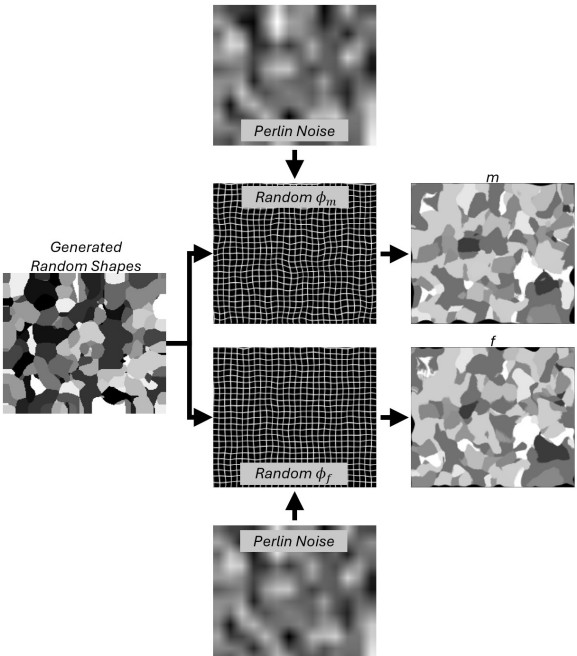

Figure 3: Process of generating random image pairs. A random image is first created using the method shown in Figure 2. Two random diffeomorphic deformation fields are then generated using three-channel Perlin noise and converted into smooth velocity fields via scaling-and-squaring (Ashburner, 2007). These deformation fields are applied to the same random image to produce an image pair for pretraining.

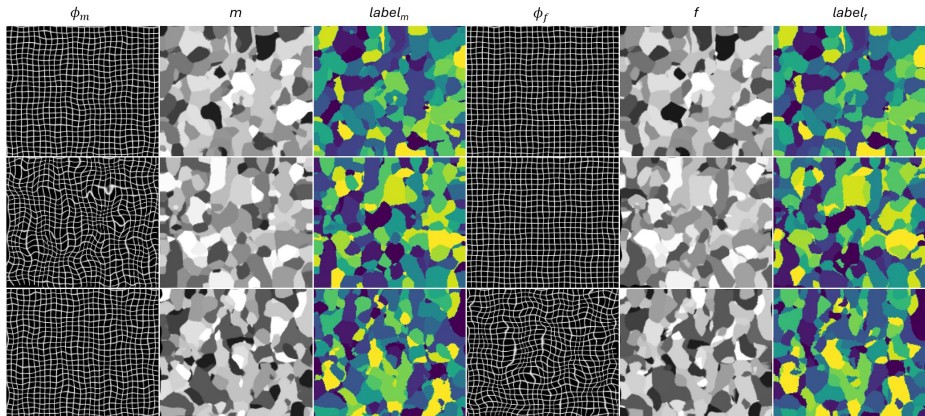

Figure 4: Visualization of three examples of randomly generated image pairs, along with the corresponding deformation fields used to create them and their associated label maps.

## Appendix D. Ablation Study Results

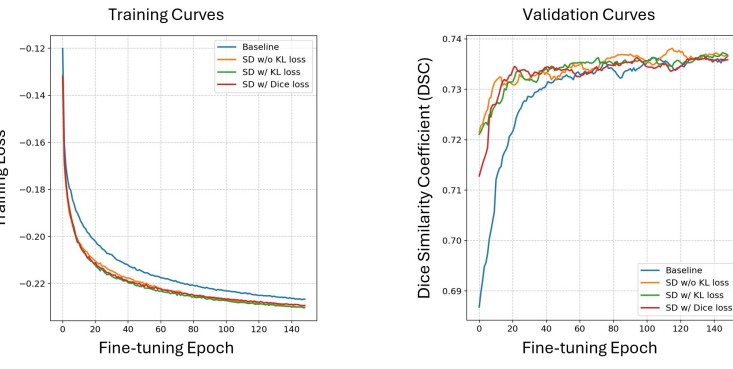

Figure 5: Training and smoothed validation curves for fine-tuning the downstream registration task in ablation studies on self-distillation. "Baseline" refers to training TransMorph from scratch without the proposed pretraining strategy. "SD w/o KL loss" uses the self-distillation pretraining scheme with $\eta = 0$ in Equation (1). "SD w/ KL loss" sets $\eta = 1e-7$ in Equation (1). "SD w/ Dice loss" replaces NCC loss with Dice loss in Equation (1), as originally used in (Hoffmann et al., 2021).

|  | Initial | Baseline | Train w/ Rand. | SD w/o KL loss | SD w/ KL loss | SD w/ Dice loss |
|---|---|---|---|---|---|---|
| **Dice**↑ | 0.386±0.195 | 0.749±0.125 | 0.653±0.130 | 0.751±0.124 | 0.751±0.122 | 0.749±0.128 |
| **%NDV**↓ | 0.00±0.00 | 0.00±0.00 | 0.00±0.00 | 0.00±0.00 | 0.00±0.00 | 0.00±0.00 |

Table 1: Quantitative results on the test set from ablation studies evaluating the impact of self-distillation. "`Baseline`" refers to training TransMorph from scratch without the proposed pretraining strategy. "`Train w/ Rand.`" refers training TransMorph from scratch using only random images. "`SD w/o KL loss`" uses the self-distillation pretraining scheme with $\eta = 0$ in Equation (1). "`SD w/ KL loss`" sets $\eta = 1e - 7$ in Equation (1). "`SD w/ Dice loss`" replaces NCC loss with Dice loss in Equation (1), as originally used in (Hoffmann et al., 2021). All models with pretraining significantly outperform the `Baseline` ($p < 0.05$, Wilcoxon signed-rank test).

## Appendix E. Results on Reducing Amount of Training Data

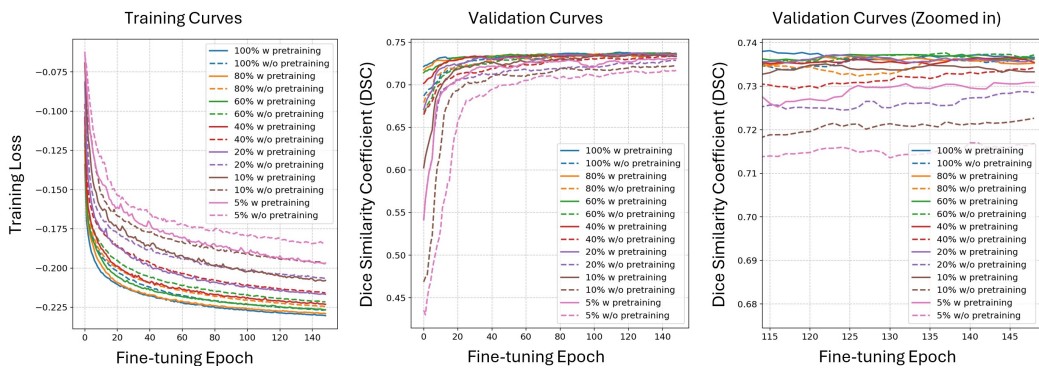

Figure 6: Visualization of training and validation curves for fine-tuning the downstream registration task, comparing models with pretraining (solid lines) and without pretraining (dashed lines). Pretraining consistently yields lower training loss and higher validation Dice scores than training from scratch, given the same amount of training data.

| | 100% | 80% | 60% | 40% | 20% | 10% | 5% |
|---|---|---|---|---|---|---|---|
| **With Pretraining** | 0.751±0.124 | 0.750±0.127 | 0.750±0.125 | 0.750±0.128 | 0.751±0.123 | 0.747±0.129 | 0.745±0.129 |
| **W/o Pretraining** | 0.749±0.125 | 0.750±0.125 | 0.750±0.124 | 0.749±0.131 | 0.742±0.130 | 0.737±0.132 | 0.733±0.132 |

Table 2: Quantitative results on the test set for models trained with and without the proposed pretraining strategy, using progressively reduced training data from 100% (403 images) to 5% (20 images). The results show that the proposed strategy enables competitive performance even with substantially less data. For example, with only 10% of the training data, the pretrained model achieves comparable performance to the non-pretrained model trained on the full dataset (0.747 vs. 0.749), while with a statistically significant difference ($p = 0.026$, Wilcoxon signed-rank test).

## Appendix F. Comparison with Pretraining with In-domain Data

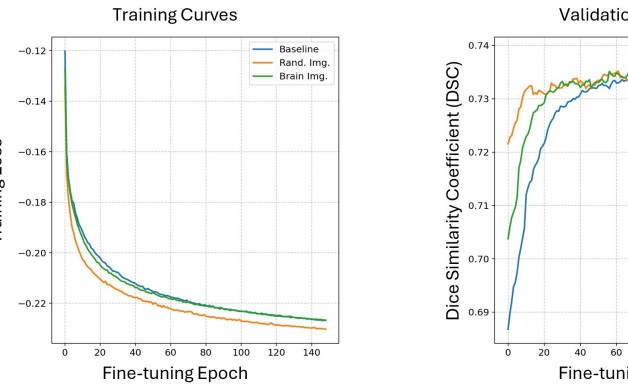

Figure 7: Visualization of training and smoothed validation curves for fine-tuning the downstream registration task, comparing models trained without pretraining (blue), pretrained with random images (orange), and pretrained with in-domain brain images (green).

|  | Initial | Baseline | Pretrain w/ Brain | Pretrain w/ Rand. |
|---|---|---|---|---|
| **Dice↑** | 0.386±0.195 | 0.749±0.125 | 0.753±0.124 | 0.751±0.124 |
| **%NDV↓** | 0.00±0.00 | 0.00±0.00 | 0.00±0.00 | 0.00±0.00 |

Table 3: Quantitative results on the test set comparing no pretraining ("`Baseline`"), pretraining with random images ("`Pretrain w/ Rand.`"), and pretraining with in-domain data ("`Pretrain w/ Brain`", i.e., brain images from a different dataset). Both pretraining strategies significantly outperform the baseline ($p < 0.05$, Wilcoxon signed-rank test).

## Appendix G. Comparison with Existing Registration Models

|  | Initial | deedsBCV | ConvexAdam | SynthMorph | TransMorph | TransMorph w/ pretraining |
|---|---|---|---|---|---|---|
| **Dice↑** | 0.386±0.195 | 0.740±0.127 | 0.749±0.126 | 0.688±0.152 | 0.749±0.125 | 0.751±0.124 |
| **%NDV↓** | 0.00±0.00 | 0.02±0.05 | 0.01±0.01 | 0.00±0.00 | 0.00±0.00 | 0.00±0.00 |

Table 4: Quantitative results on the test set comparing the proposed method with existing registration models.

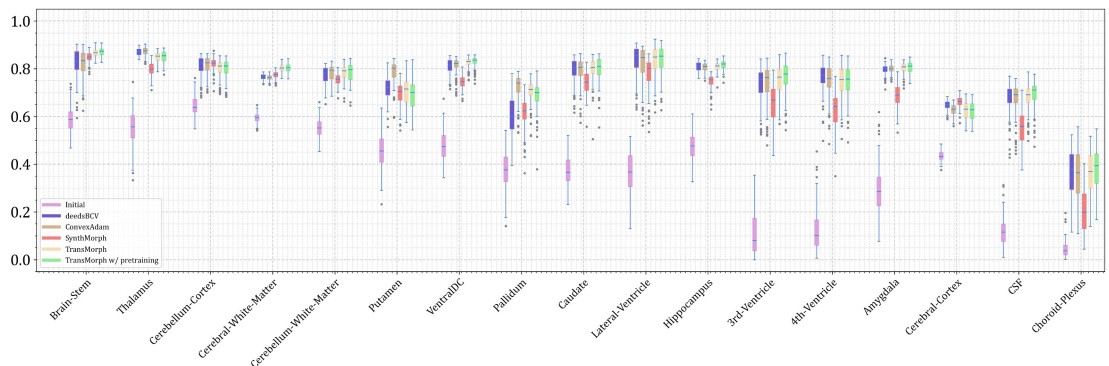

Figure 8: Detailed quantitative results across various brain structures for different registration methods.

## Appendix H. Application to a Data-limited Scenario

To further demonstrate the proposed method, we applied it to a highly data-limited setting: lung registration using the 4DCT dataset (Castillo et al., 2009a,b), which contains only 10 subjects. Due to variations in image size and resolution, all images were resampled and cropped or zero-padded to fit the lung region within a standardized spatial dimension of

$224 \times 224 \times 224$. Specifically, Case 1 was resampled to a voxel size of $1.14 \times 1.14 \times 1.14$ mm, Cases 2–5 to $1.4 \times 1.4 \times 1.4$ mm, and Cases 6–10 to $1.6 \times 1.6 \times 1.6$ mm. Landmarks were adjusted by defining a 2.5-voxel-radius sphere around each original point, and using the centroid of the transformed sphere after resampling and cropping as the updated landmark position. Given the limited dataset, we selected only four subjects (Subjects 7–10) for training, and the remaining six (Subjects 1–6) for validation. The models were trained for 400 epochs using the Adam optimizer (learning rate = 0.0001) with random flipping along a random axis for data augmentation. Registration performance was evaluated using the targeted registration error (TRE) between landmarks of the deformed and fixed images.

Figure 9 shows the training and validation curves for TransMorph with and without pretraining. Consistent with the results of the IXI dataset, pretraining reduces training loss and improves validation performance, achieving lower TRE and more stable validation curves. Detailed TRE results for each validation case are provided in Table 5. Although the pretrained model significantly outperforms TransMorph trained from scratch, it remains suboptimal compared to existing pairwise optimization-based methods (i.e., deedsBCV and ConvexAdam). However, given that our approach was trained using only four image pairs, achieving TRE values within two voxel sizes is still noteworthy. We anticipate that incorporating instance-specific optimization for each case, initialized by our pretrained model, could further improve registration accuracy.

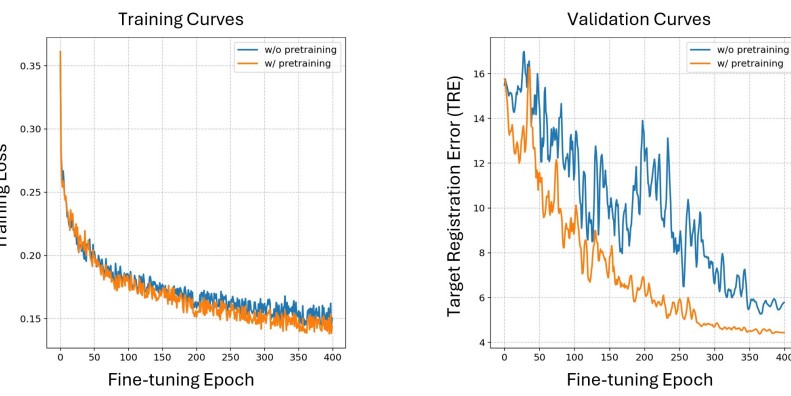

Figure 9: Training and smoothed validation curves for lung registration on the 4DCT dataset.

| Case No. | Initial | deedsBCV | ConvexAdam | TransMorph | TransMorph w/ pretraining |
|---|---|---|---|---|---|
| **1** | 3.938±2.809 | 1.079±0.646 | 1.073±0.576 | 2.721±1.511 | 1.420±0.912 |
| **2** | 4.366±3.931 | 1.024±0.679 | 1.027±0.658 | 1.858±1.377 | 1.306±0.941 |
| **3** | 6.977±4.100 | 1.349±0.825 | 1.257±0.702 | 2.690±2.279 | 2.761±2.654 |
| **4** | 9.906±4.882 | 1.543±1.003 | 1.483±1.004 | 2.515±1.804 | 2.108±1.419 |
| **5** | 7.525±5.540 | 1.621±1.479 | 1.503±1.412 | 2.851±2.289 | 2.275±2.044 |
| **6** | 10.957±7.017 | 1.898±1.404 | 1.686±1.332 | 3.884±2.914 | 2.824±2.003 |
| Avg. TRE↓ | 7.278±4.713 | 1.419±1.006 | 1.338±0.947 | 2.753±2.029 | 2.116±1.662 |
| %NDV↓ | 0.00% | 0.00% | 0.00% | 0.00% | 0.00% |

Table 5: Quantitative results on the validation set of 4DCT dataset comparing the proposed method with existing registration models.

