# OpenReview forum: "Pretraining Deformable Image Registration Networks with Random Images"
_MIDL.io/2025/Short_Papers — MIDL 2025 - Short Papers_

### Official Review · Reviewer_HrWB · 2025-04-28

**Rating:** 5
**Confidence:** 5

**Summary:**

This paper presents a pre-training strategy for deformable medical image registration networks using random images instead of real medical datasets. The authors demonstrate the effectiveness of the proposed method through comparisons with several existing approaches, both with and without the use of pre-training.

**Strengths:**

- Using image registration on random data as a pretraining to reduce the cost of medical data curation is an interesting and practical approach. This work would make a valuable discussion to MIDL.

- The low-data experiments on brain MRI and lung CT datasets are particularly appealing, demonstrating substantial performance gains using only 5%–10% of the data.

- The paper is well written and well organized, making it easy for readers to follow and understand the methodology and results.

**Weaknesses:**

- It would be interesting to see the model evaluated on a broader range of datasets, such as abdominal, cardiac, or multi-modal imaging, to further strengthen the generalizability claims.

- While the paper compares favorably against SynthMorph and other registration methods, it would also be interesting to include direct comparisons to more recent models targeting foundation model development, such as [1,2].

[1] A Foundation Model for Medical Image Registration, 2024.
[2] UniReg: Foundation Model for Controllable Medical Image Registration, 2025.

However, given the space limitations of a short paper, it is understandable that these experiments can not be included; it would be interesting to see them explored in an extended version or future work.

---

### Decision · Program_Chairs · 2025-05-01

Accept